# Collagen Mimetic Peptides Promote Repair of MMP-1-Damaged Collagen in the Rodent Sclera and Optic Nerve Head

**DOI:** 10.3390/ijms242317031

**Published:** 2023-12-01

**Authors:** Ghazi O. Bou Ghanem, Dmitry Koktysh, Robert O. Baratta, Brian J. Del Buono, Eric Schlumpf, Lauren K. Wareham, David J. Calkins

**Affiliations:** 1Vanderbilt Eye Institute, Department of Ophthalmology and Visual Sciences, Vanderbilt University Medical Center, Nashville, TN 37232, USA; ghazi.b.ghanem@vumc.org; 2Vanderbilt Institute of Nanoscale Science and Engineering, Vanderbilt University, Nashville, TN 37212, USA; 3Stuart Therapeutics, Inc., Stuart, FL 34994, USA; bob@stuarttherapeutics.com (R.O.B.); eric@stuarttherapeutics.com (E.S.)

**Keywords:** tissue biomechanics, extracellular matrix, extracellular matrix remodeling, collagen, collagen mimetic peptides

## Abstract

The structural and biomechanical properties of collagen-rich ocular tissues, such as the sclera, are integral to ocular function. The degradation of collagen in such tissues is associated with debilitating ophthalmic diseases such as glaucoma and myopia, which often lead to visual impairment. Collagen mimetic peptides (CMPs) have emerged as an effective treatment to repair damaged collagen in tissues of the optic projection, such as the retina and optic nerve. In this study, we used atomic force microscopy (AFM) to assess the potential of CMPs in restoring tissue stiffness in the optic nerve head (ONH), including the peripapillary sclera (PPS) and the glial lamina. Using rat ONH tissue sections, we induced collagen damage with MMP-1, followed by treatment with CMP-3 or vehicle. MMP-1 significantly reduced the Young’s modulus of both the PPS and the glial lamina, indicating tissue softening. Subsequent CMP-3 treatment partially restored tissue stiffness in both the PPS and the glial lamina. Immunohistochemical analyses revealed reduced collagen fragmentation after MMP-1 digestion in CMP-3-treated tissues compared to vehicle controls. In summary, these results demonstrate the potential of CMPs to restore collagen stiffness and structure in ONH tissues following enzymatic damage. CMPs may offer a promising therapeutic avenue for preserving vision in ocular disorders involving collagen remodeling and degradation.

## 1. Introduction

Collagen-rich ocular tissues such as the cornea and sclera function as a robust biomechanical framework to maintain eye structure and integrity, an important function achieved through the laminar arrangement of collagen [1]. Degradation of collagen in the sclera and scleral remodeling is linked to the progression of myopia, a prominent driver of visual impairment worldwide [2,3,4,5]. High myopia heightens the risk of sight-threatening complications such as myopic macular degeneration and retinal detachment [6]. Scleral remodeling and thinning, tissue degradation, and change in collagen fiber architecture underlie axial elongation in myopia [7,8,9]. Similarly, in glaucoma, the leading cause of irreversible blindness worldwide [10,11], the organization of collagen fibers in the peripapillary sclera (PPS) is altered in human patients and in animal models. Remodeling of the extracellular matrix (ECM)—including collagen—also occurs at the optic nerve head (ONH), a critical juncture in glaucoma pathophysiology [12,13]. Collagen fiber density is diminished in glaucomatous lamina cribrosa [14], changing the biological and biomechanical properties of the tissue to impact retinal ganglion cell (RGC) vulnerability at ONH [15,16,17,18,19].

An interdependence exists between the sclera and lamina cribrosa; collagen fibers of the PPS intersect with collagen fibers at the border of the lamina cribrosa [20,21], together forming the ONH connective tissue framework [22]. In rodents the glial lamina, which is analogous to the lamina cribrosa in primates, maintains the structural integrity of the ONH, despite having a less extensively structured collagenous ECM [14,23]. Matrix metalloproteinases (MMPs) are proteolytic enzymes responsible for ECM degradation, and play an important role in collagen remodeling in ocular tissues during both homeostasis and disease [3,24]. A large component of scleral and ONH tissue is collagen, which imparts stiffness and structural integrity to tissue, as well as mediating important inflammatory and cellular signaling [22,25,26,27,28]. Atomic force microscopy (AFM) offers a potent approach to assess the biomechanical properties of tissue, including stiffness, without the need for extensive tissue processing [29]. Using AFM, the Young’s modulus, a mechanical property that is indicative of tissue stiffness, can be determined quantitatively [30]. In the eye, AFM has been successfully used to evaluate stiffness across various ocular tissues and in a broad range of mammals, including corneal stiffness in mice [31] and humans [32,33]; trabecular meshwork stiffness in rats [34]; lens stiffness in mice [35] and non-human primates [36]; and the sclera and optic nerve head stiffness in pigs [37,38], mice [39], rats [40], and humans [41]. 

Our previous work has highlighted the capacity of collagen mimetic peptides (CMPs) to repair damage to tissue in the optic projection [42]. CMPs also promote corneal epithelium healing and epithelial cell regeneration post acute injury [43], and corneal nerve repair in a dry eye model [44]. A signature characteristic of intact collagen is a triple helical structure, which contains individual triple helices called tropocollagens [45]. Tropocollagens include a set of three polypeptide chains comprising repeating sequences of glycine-x-y triplets (x and y most commonly represent proline and hydroxyproline) [45]. CMPs are short, single-stranded peptides that bind with high avidity to damaged collagen while bypassing intact collagen structures [46,47,48]. CMPs enhance wound closure in mice through improved collagen fibril alignment after damage by MMP [43,47]. The PPS and the glial laminal tissue by design contain large amounts of collagen-rich ECM. Thus, CMPs are largely sequestered by damaged collagen contained within the ECM. Given the established capacity of CMPs to reinstate the structural and functional attributes of collagen to their native states, here, we sought to determine the potential of CMPs in restoring the biomechanical properties of scleral and ONH tissue after enzymatic treatment. Using AFM, we measured rat PPS and glial lamina stiffness at baseline, followed by MMP-1 treatment, and again after CMP-3 or 1x PBS (vehicle) treatment. We found that MMP-1 effectively reduces stiffness and increases levels of fragmented collagen in the PPS and glial lamina. Application of CMP after MMP-1 treatment partially restores stiffness and reduces collagen fragmentation compared to vehicle controls. Our results suggest that CMP partially restores the structural and biomechanical properties of damaged collagen in scleral and glial lamina tissue and may therefore represent a novel therapeutic avenue in ocular diseases where collagen remodeling and degradation occur.

## 2. Results

### 2.1. Collagen Mimetic Peptide Restores Stiffness of the PPS after MMP-1

To determine tissue stiffness in the PPS, AFM measurements were acquired from distinct PPS locations within 300–500 µm from the ONH. Measurements were taken at baseline in naïve tissue, and in the same location after MMP-1 incubation and EDTA quenching, and subsequent CMP-3 or vehicle (1x PBS) treatments. Representative experimental datasets for a specific location are shown in Figure 1A,B. Incubation with MMP-1 decreased the average Young’s modulus by 38.3% (Figure 1C, Table 1), indicating diminished tissue stiffness. CMP-3 reversed this trend, increasing the average Young’s modulus by 23.0% (Figure 1C, Table 1), signifying at least a partial restoration of PPS stiffness. Application of vehicle after MMP-1 treatment led to a further 26.0% reduction in average Young’s modulus, or a total reduction of 54.3% from baseline (Figure 1C, Table 1).

### 2.2. Collagen Mimetic Peptide Restores Stiffness of the Glial Lamina after MMP-1

Glial lamina stiffness measurements were determined as outlined for the PPS. Representative experimental data are shown in Figure 2A,B. The average Young’s modulus of the glial lamina decreased by 57.2% after MMP-1 treatment; CMP-3 again partially restored stiffness of the tissue (49.5% increase; Figure 2C, Table 2). Although stiffness increased by 13.2% with application of the vehicle (Figure 2C, Table 2), it did not increase the stiffness of the tissue to the same magnitude as observed with CMP-3.

### 2.3. Collagen Mimetic Peptide Mitigates Collagen Fragmentation

To assess collagen fragmentation, we immunolabeled rat ONH tissue after AFM analysis and imaged the glial lamina (Figure 3A; box 1) or PPS regions of interest (Figure 3A; box 2). Collagen type-1 is visualized in green, with fragmented collagen highlighted in red using an R-CHP peptide; an increase in red fluorescence is indicative of increased fragmented collagen. In naïve glial lamina tissue at baseline, collagen-1 was evident with low levels of fragmentation (Figure 3B(1a)). After MMP-1 digestion and vehicle addition, the level of fragmented collagen increased dramatically (Figure 3B(1b)). After collagen digestion with MMP-1 and subsequent CMP-3 addition, the level of fragmented collagen was reduced and comparable to the baseline (Figure 3B(1c)). Similarly, in the PPS, levels of fragmented collagen were low at baseline (Figure 3B(2a)). After MMP-1 and vehicle addition, fragmented collagen levels increased (Figure 3B(2b)). After CMP-3 addition to MMP-1 digested tissue, the levels of fragmented collagen were again reduced (Figure 3B(2c)).

## 3. Discussion

Scleral and ONH tissue integrity is critical to the maintenance of eye structure and the support of RGC axons as they traverse the ONH and travel to the brain. Breakdown of ONH and scleral tissue integrity through damage to collagen is evident in diseases that lead to vision loss, including myopia and glaucoma [7,9,14,17]. Collagen degradation with aging and disease is often attributed to increases in tissue MMP activity that promotes collagen breakdown and triggers tissue remodeling [3,24]. Our work with CMPs has demonstrated a broad capacity for their use in tissue repair of the eye and optic projection. In this study, we sought to explore whether CMPs had the capacity to directly repair damaged collagen in scleral and glial lamina tissues. To do this, we measured tissue stiffness using AFM and visualized fragmented collagen using immunohistochemistry. 

Our results here show that degradation of scleral and ONH tissue ex vivo by MMP-1 effectively decreased tissue stiffness as measured by AFM (Figure 1 and Figure 2). This was expected, since proteolytic cleavage of collagen type I fibrils by MMP-1 reduces stiffness in vitro [49], and enzymatic treatment of porcine sclera and cornea reduces tissue stiffness and collagen organization [37,50]. Our results also show that a reduction in glial lamina and scleral tissue stiffness after MMP-1 addition coincides with increased binding of fragmented collagen, as detected by increased fluorescence in such tissues due to the presence of a collagen peptide (R-CHP) that binds specifically to fragmented collagen (Figure 3). These results suggest that the reduction in stiffness observed after MMP-1 was indeed due to collagen fragmentation in the tissue. Application of CMP-3 to tissue after MMP-1 degradation partially restored scleral and glial lamina stiffness (Figure 1 and Figure 2, and Table 1 and Table 2). Interestingly, the addition of vehicle (1x PBS) after MMP-1 led to a further reduction in the Young’s modulus of the sclera, possibly due to intrinsic MMP activity or a lack of quenching of the MMP-1 reaction by EDTA (Figure 1, Table 1). In the glial lamina, however, the addition of vehicle led to a slight increase in tissue stiffness (Figure 2, Table 2). The biological composition of scleral and glial lamina tissues is distinct, so it is not unexpected that they may behave differently under experimental conditions. Nevertheless, the addition of CMP-3 partially restored tissue stiffness in both the glial lamina and PPS, and such stiffening was associated with a reduction in the level of fragmented collagen in the tissue (Figure 3). CMP-3 has a high affinity for damaged collagen, intercalating into collagen strand breaks, repairing the native triple helical structure, and reducing collagen strand breaks or fragmentation directly [48]. Repairing collagen structure may also have important implications for downstream immune signaling, which could in turn alter the expression and activation of tissue-resident MMP enzymes to impact collagen stiffness and levels of fragmentation [51,52,53]. 

Our demonstration that CMPs have the potential to stiffen collagen-rich ocular tissues could have important implications for ocular diseases that are characterized in part by collagen degradation and tissue remodeling. For instance, in myopia, increases in scleral MMP activity accelerates scleral remodeling, leading to scleral thinning and progression of the disease [54]. In animal models of myopia, preventing collagen damage with TIMP-2 reduced scleral collagen degradation and development of myopia [55]. Myopia is also associated with decreased collagen-1 expression [56] and inhibition of collagen crosslinking accelerated myopia development [57]. Furthermore, sub-tenon injections of genipin to increase scleral crosslinking in a guinea pig model of myopia prevented myopia progression [58]. Similarly, glaucoma is associated with ONH tissue alterations that challenge the health of the optic nerve [22]. Interestingly, the incidence of myopia increases the risk of glaucoma independently of intraocular pressure (IOP) and other risk factors [59]. Collagen is the predominant component of both scleral and ONH tissue. There is evidence that scleral and lamina cribrosa stiffness increases with age and with glaucoma [49,60]. Scleral crosslinking in mice using glutaraldehyde alters the pressure–strain relationship of tissue at the ONH, leading to nerve degeneration [61]. However, it is argued that reduced strain at the lamina cribrosa is potentially protective to the nerve [62,63]. Furthermore, elsewhere in the CNS (cerebral cortex and spinal cord), ECM tissue softens after injury [51]. 

In summary, our results suggest that CMP-3 has the capacity to increase the stiffness of ONH tissue and reduce levels of collagen fragmentation after MMP-1 digestion. These results hint at a possible therapeutic role for CMPs in myopia or glaucoma. An understanding of the involvement of tissue stiffness in ocular diseases that lead to visual impairment, including glaucoma and myopia, is required for developing new treatments that address changes in the pliability of collagen. Since collagen damage in tissue may be evident early in these diseases, repair of damaged collagen using mimetic peptides may be an effective preventative therapeutic approach. 

## 4. Materials and Methods

### 4.1. Animals

For all experiments, male Brown Norway rats (*n* = 6; 3 rats for PPS measurements and 3 rats for glial lamina measurements) aged 3 months were obtained from Charles River Laboratories (Wilmington, MA, USA). This study was conducted in accordance with the ARVO Statement for the Use of Animals in Ophthalmic and Vision Research. Animal protocols were approved by the Institutional Animal Care and Use Committee of the Vanderbilt University Medical Center. Rats were housed in a facility managed by the Vanderbilt University Division of Animal Care, with ad libidum access to water and standard rat chow and a 12 h light cycle (lights on at 6:30 a.m. and off at 6:30 p.m.).

### 4.2. Tissue Preparation

Rats were anesthetized with isoflurane before decapitation. Both eyeballs were rapidly enucleated and placed in ice-cold 1x phosphate-buffered saline (PBS). Eyes were bisected posterior to the equator, the anterior segment and lens were removed, and ONH tissue was embedded in an Optimal Cutting Temperature medium (OCT, Fisher Healthcare, reference # 4585) for cryo-sectioning. Sagittal eye sections through the ONH were cut at 20 µm thickness and were mounted on Poly-D-Lysine (PDL; Cat no: A38904-01–ThermoFisher Scientific, Frederick, MD, USA) coated glass coverslips. Samples were stored on dry ice until same-day AFM imaging. 

### 4.3. Atomic Force Microscopy

ONH tissue sections on coverslips were mounted onto the AFM equipment so that the PPS and glial lamina were clearly in view (Figure 4). Tissue stiffness was acquired using PeakForce Quantitative Nanomechanical Mapping (QNM) in Fluid AFM imaging mode (Bruker, Santa Barbara, CA, USA). A SAA-SPH-5UM probe (Bruker, Santa Barbara, CA, USA) with a 5 μm end radius and a 0.25 N/m nominal spring constant was used to indent the PPS tissue to measure PPS stiffness. A CP-PNPL-SiO-D probe (Bruker, Santa Barbara, CA, USA) with a 5 µm tip radius and 0.08 N/m spring constant was used to measure glial lamina stiffness. Force–displacement curves were fit to the Hertz model assuming a Poisson’s ratio of 0.5 using the Bruker curve fitting software to determine the elastic modulus (Young’s modulus). Before data collection, the probe was calibrated in liquid using the thermal tune method with the addition of 1x PBS [64]. In each tissue location, force volume maps were acquired using a scan size of 10 µm, with 16 samples/line for a total of 256 force–displacement curves, at a scan rate of 1 Hz. Up to 6 distinct PPS locations approximately 300–500 µm away from the edge of the ONH were used for baseline measurements in each sample, and up to 5 distinct locations per sample were taken as baseline measurements in the glial lamina. After MMP-1 and CMP-3 or 1x PBS treatment, the same locations were remeasured to determine changes in tissue stiffness.

### 4.4. Matrix Metalloproteinase Treatment

Recombinant human MMP-1 (Biolegend, catalog # 592902), was used to degrade tissue collagen at a final concentration of 0.05 mg/mL (sclera) or 0.005 mg/mL (glial lamina) in 1x PBS. For PPS stiffness measurements, ONH tissue was treated with 15 µL MMP-1 (0.05 mg/mL) for 30 min at 37 °C. The sample was washed with 1x PBS before the reaction was quenched with 75 µL of 0.1 M EDTA (Invitrogen, reference # 15575-038) in PBS for 15 min at room temperature. For glial lamina stiffness measurements, 15 µL MMP-1 (0.005 mg/mL) was added for 20 min at 37 °C, washed with 1x PBS and quenched with 75 µL of 0.1 M EDTA in PBS for 15 min at room temperature. Finally, samples were washed with 1x PBS and AFM data was collected.

### 4.5. Collagen Mimetic Peptide or Vehicle Treatment

After measuring tissue stiffness post-MMP-1 treatment, 75 µL of CMP-3 at a final concentration of 200 µM or an equivalent volume of 1x PBS (vehicle) was added for 60 min at 37 °C. Samples were then washed with 1x PBS, and final AFM measurements were acquired. CMP-3 is a 21-residue single-strand peptide consisting of a 7-repeat sequence of proline (Pro) and glycine (Gly) as (Pro-Pro-Gly)_7_. This structure is similar to CMPs known for their high-affinity intercalation with damaged type I collagen in vitro and in vivo [46,47,48]. It was supplied by Bachem, AG (Germany) and was produced using standard solid-phase peptide synthesis (SPPS) chemistry, followed by purification through preparative liquid chromatography on a reversed-phase column with acetonitrile (ACN) gradient elution and UV detection at 230 nm. Collected fractions were analyzed via ultra-high-performance liquid chromatography (UHPLC), pooled, and diluted with water to reduce their ACN concentration. Further purification included salt exchange and microfiltration through a 0.45 μm membrane filter, followed by lyophilization, resulting in a pre-clinical use product with a purity range of 90.5–90.7%. A flow diagram showing tissue endpoints and treatments is shown in Figure 5 for clarity.

### 4.6. Immunohistochemistry

After AFM measurements, tissues were fixed in 4% paraformaldehyde (PFA) solution for 5 min and washed with 1x PBS at room temperature. Auto-fluorescence was quenched by adding 0.1% sodium borohydride/1x PBS for 30 min at room temperature. Tissue was washed 2X for 10 min per wash in 1x PBS solution. Tissue was then blocked in a solution containing 5% normal donkey serum (NDS; 017-000-121, Jackson ImmunoResearch Laboratories, Inc., West Grove, PA, USA) and 0.1% Triton X-100/1x PBS for 1 h at room temperature. After blocking, the tissue was placed in primary antibody solution (3% NDS/0.1% Triton X-100 in 1x PBS) with 20 µm collagen-hybridizing peptide (R-CHP, Cy3 Conjugate; 3Helix, Salt Lake City, UT, USA) and 1:100 anti-mouse collagen-1 (Ab6308, Abcam, Waltham, MA, USA). Samples were covered with paraffin overnight at 4 °C protected from light. The next day, the tissue was washed 3X for 10 min per wash in 1x PBS. Tissue was then placed in a secondary antibody solution (1% NDS/0.1% Triton X-100 in 1x PBS) containing 1:400 Donkey anti-mouse Alexa Fluor-488 (715-546-150, Jackson ImmunoResearch Laboratories, Inc., West Grove, PA, USA) for 2 h at room temperature, protected from light. Tissue was then washed 3X in 1x PBS for 10 min per wash and mounted in DAPI Fluoromount-G (0100-20, SouthernBiotech, Birmingham, AL, USA) for confocal imaging. 

### 4.7. Optic Nerve Head Tissue Imaging

Fluorescent optic nerve head tissue images were taken using an Olympus FV-1000 inverted confocal microscope and a 20X or 40X objective.

### 4.8. Statistical Analysis

All data are presented as mean ± 95% C.I. unless otherwise stated. Graphs were generated, and statistical analyses were conducted using GraphPad Prism version 9.0 (GraphPad Software, San Diego, CA, USA). Normality was assessed using the Shapiro–Wilk test of normality. If the data demonstrated a normal distribution, we conducted parametric statistical analyses, such as a *t*-test and analysis of variance (ANOVA). In cases wherein the data did not follow a normal distribution, we employed non-parametric tests, i.e., the Mann–Whitney test and the Kruskal–Wallis test followed by Dunn’s multiple comparisons test, as specified in the figure legends. Statistical significance was defined as a *p*-value of 0.05 or less.

## Figures and Tables

**Figure 1 ijms-24-17031-f001:**
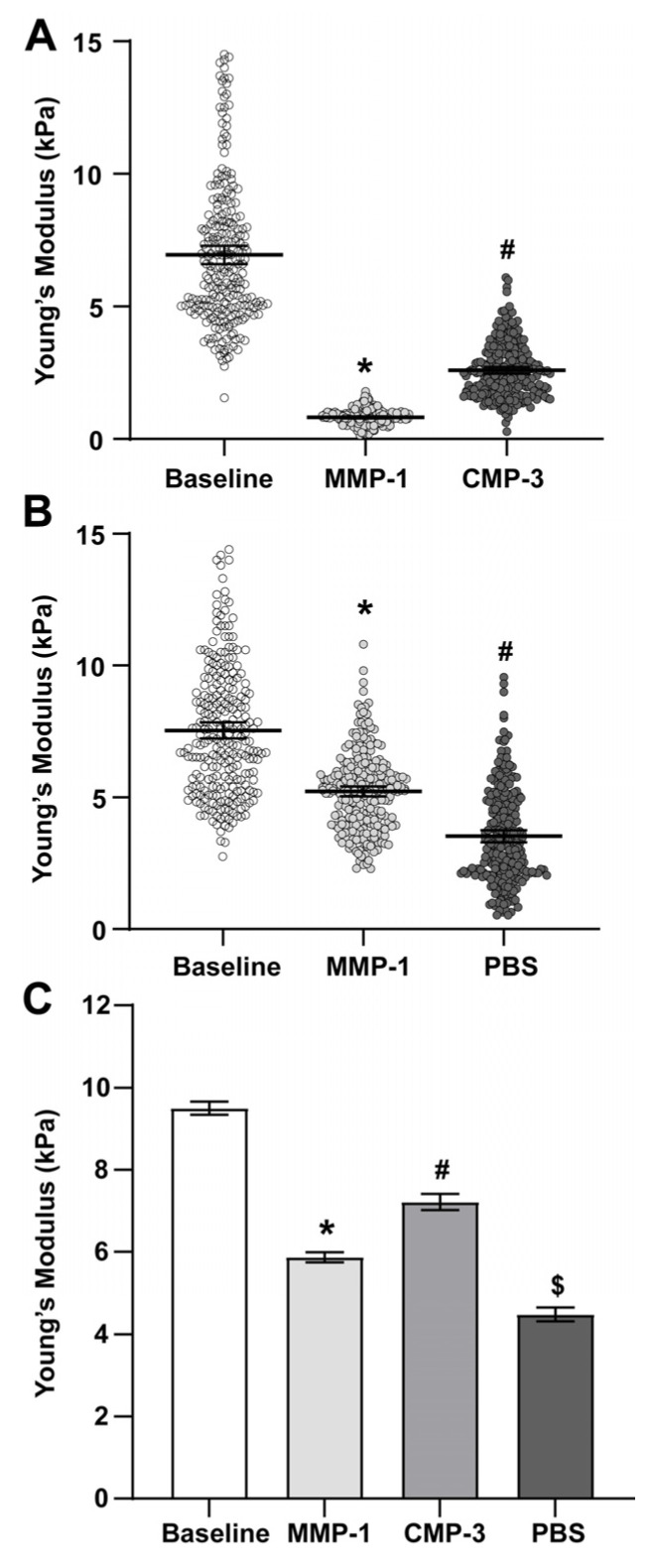
CMP-3 partially restores PPS stiffness after MMP-1 treatment. (**A**) Representative Young’s moduli measurements in PPS from *n* = 1 of 3 animals. MMP-1 treatment reduced the stiffness of the PPS (*, *p* ≤ 0.001), while CMP-3 partially restored tissue stiffness (#, *p* ≤ 0.001). (**B**) Representative Young’s moduli measurements in PPS from *n* = 1 of 3 animals. The addition of vehicle (1x PBS) after MMP-1 treatment gradually reduced tissue stiffness (*, #, *p* ≤ 0.001). (**C**) Pooled Young’s moduli mean values from *n* = 3 animals show that MMP-1 treatment significantly reduced PPS tissue stiffness compared to baseline (*, *p* ≤ 0.001). Treatment with CMP-3 after MMP-1 digestion partially restored tissue stiffness (#, *p* ≤ 0.001), while the addition of vehicle after MMP-1 treatment further reduced tissue stiffness ($, *p* ≤ 0.001).

**Figure 2 ijms-24-17031-f002:**
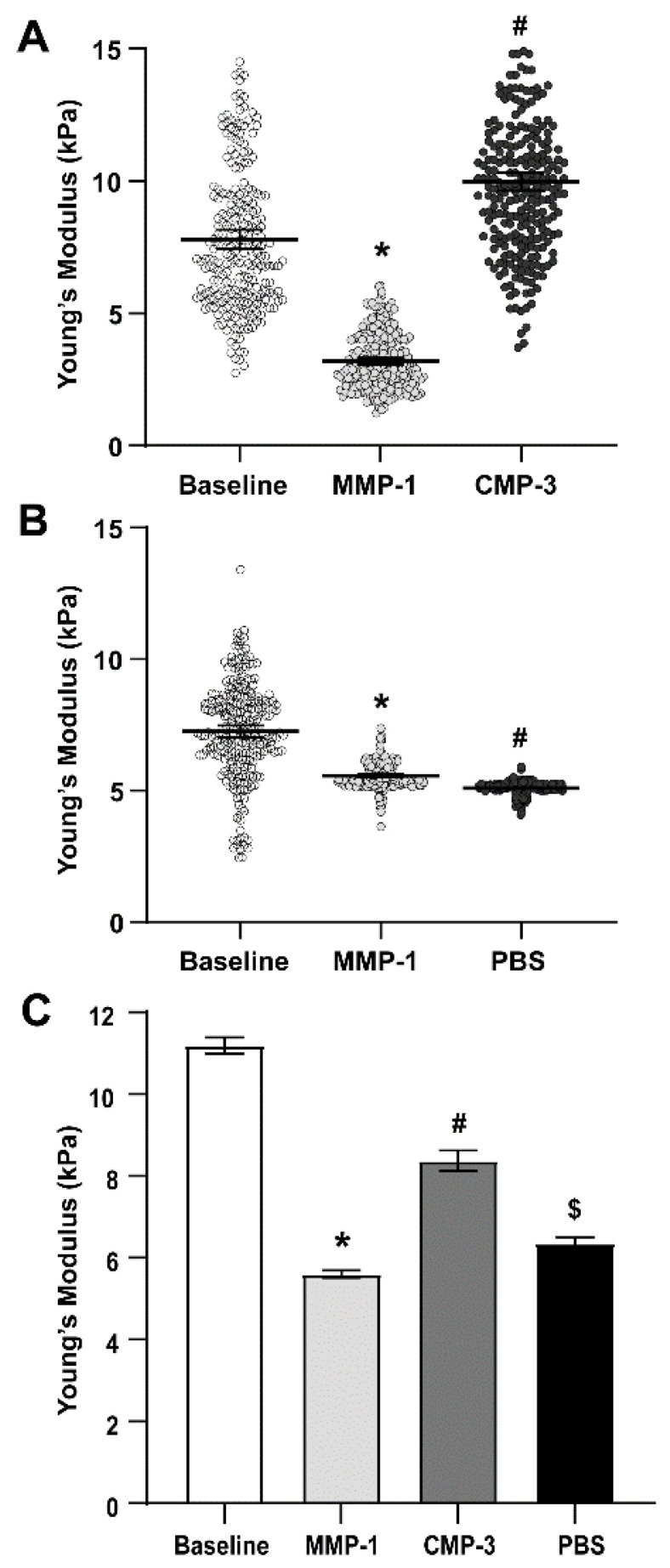
CMP-3 partially restores glial lamina stiffness after MMP-1 treatment. (**A**) Representative Young’s moduli measurements in glial lamina from *n* = 1 of 3 animals. MMP-1 treatment reduced stiffness of the PPS (*, *p* ≤ 0.001), while CMP-3 restored tissue stiffness (#, *p* ≤ 0.001). (**B**) Representative Young’s moduli measurements in glial lamina from *n* = 1 of 3 animals. The addition of vehicle (1x PBS) after MMP-1 treatment gradually reduced tissue stiffness (*, #, *p* ≤ 0.001). (**C**) Pooled Young’s moduli mean values from *n* = 3 animals show that MMP-1 treatment significantly reduced glial lamina tissue stiffness compared to baseline (*, *p* ≤ 0.001). Treatment with CMP-3 after MMP-1 digestion partially restored tissue stiffness (#, *p* ≤ 0.001), while the addition of vehicle after MMP-1 treatment increased tissue stiffness to a smaller extent ($, *p* ≤ 0.001).

**Figure 3 ijms-24-17031-f003:**
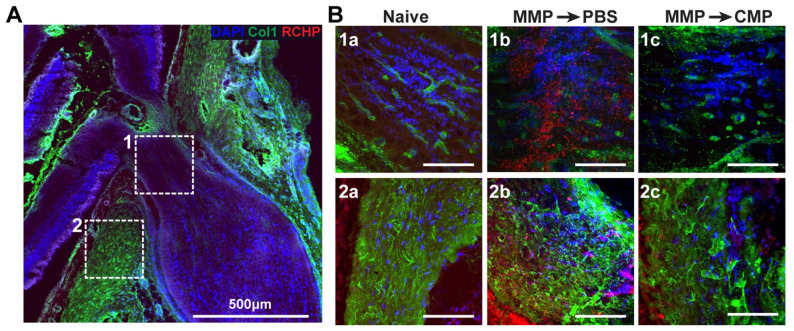
CMP repairs fragmented collagen in PPS and glial lamina. (**A**) Representative confocal image of rat ONH tissue labeled for collagen-1 (green) and fragmented collagen (R-CHP; red). Dashed boxes show regions of interest including (1) glial lamina and (2) PPS. (**B**) Representative confocal images of ROIs from glial lamina (**1a**–**1c**) and PPS (**2a**–**2c**). In naïve glial lamina, low levels of fragmented collagen were observed (**B**(**1a**)). After MMP-1 digestion, collagen fragmentation increased (**B**(**1b**)). CMP-3 treatment of MMP-1-degraded tissues reduced fragmented collagen levels (**B**(**1c**)). In naïve PPS, there were low levels of fragmented collagen (**B**(**2a**)). After MMP-1 digestion, increased fragmented collagen was evident (**B**(**2b**)). CMP-3 treatment of MMP-1-degraded tissues reduced fragmented collagen levels (**B**(**2c**)).

**Figure 4 ijms-24-17031-f004:**
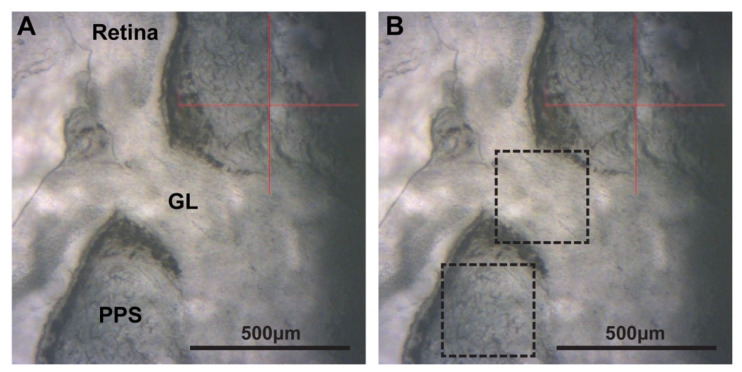
AFM microscope view. (**A**) Rat ONH cryosection with major tissue landmarks including retina, glial lamina (GL), and peripapillary sclera (PPS) indicated. (**B**) ROIs outlined with dashed boxes demonstrate where Young’s moduli measurements were taken. Scale bars as indicated.

**Figure 5 ijms-24-17031-f005:**
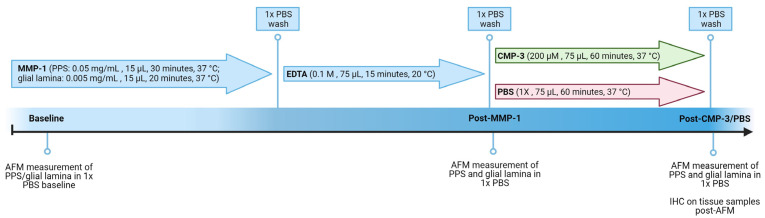
Schematic outlining the study experimental design. Tissue stiffness measurements are acquired at baseline, after 20 or 30 min of matrix metalloprotease-1 (MMP-1) treatment and 15 min of quenching with EDTA, and after 60 min of collagen mimetic peptide-3 (CMP-3) or vehicle (1x PBS) treatment.

**Table 1 ijms-24-17031-t001:** Young’s moduli values for PPS. Mean Young’s moduli values for PPS along with 95% confidence interval. *n* indicates the number of individual AFM measurements from a total of 3 rats.

Sample Type	Mean Young’s Modulus (kPa)	95% C.I.	*n*
Baseline	9.500	0.161	6656
MMP-1	5.865	0.124	5120
CMP-3	7.214	0.101	2938
1x PBS (vehicle)	4.345	0.323	1536

**Table 2 ijms-24-17031-t002:** Young’s moduli values for glial lamina. Mean Young’s moduli values for PPS along with 95% confidence interval. *n* indicates the number of individual AFM measurements from a total of three rats.

Sample Type	Mean Young’s Modulus (kPa)	95% C.I.	*n*
Baseline	13.096	0.248	9082
MMP-1	5.602	0.090	2304
CMP-3	8.373	0.253	2229
1x PBS (vehicle)	6.342	0.151	1024

## Data Availability

The data presented in this study are available on request from the corresponding authors.

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
