# Peer review of "Collagen Mimetic Peptides Promote Repair of MMP-1-Damaged Collagen in the Rodent Sclera and Optic Nerve Head"

_ijms, 2023, doi:10.3390/ijms242317031_

Round 1
Reviewer 1 Report
Comments and Suggestions for Authors
Ghanem et al. employed Atomic Force Microscopy to explore the capacity of collagen mimetic peptides (CMPs) in restoring stiffness in ocular tissues. The findings revealed that CMP-3 enhances collagen stiffness and diminishes collagen fragmentation. The manuscript maintains clarity throughout; however, pivotal information concerning the lead peptide, CMP-3, seems completely absent. Additionally, a deeper investigation into the mechanisms by which CMP-3 curtails collagen fragmentation is also essential. Here are specific comments and suggestions:
1. The current version of the manuscript lacks references numbered 34 through 62.
2. A richer background on collagen mimetic peptides (CMPs) is needed. It would be beneficial to introduce their origins, offer a clear definition, and delineate their structural comparisons with collagen. Furthermore, a detailed exposition on the sequence, structure, and physicochemical attributes of CMP-3 would greatly enhance understanding.
3. The Materials section appears incomplete without a list of materials and reagents utilized. Most crucially, the source and purity level of CMP-3 should be explicitly stated.
4. The y-axis labeling in Figure 1B is absent.
5. The manuscript would benefit from a clarification regarding the localization of CMP-3. Is it primarily internalized within the tissues, or does it predominantly inhabit the extracellular matrix (ECM)?
6. An elucidation on the mechanism of CMP-3 in attenuating collagen fragmentation is necessary. Specifically, is there any interaction between CMP-3 and MMP-1?
Author Response
We thank the reviewers of our manuscript for their thoughtful and comprehensive comments. We are delighted to submit our revision, in which we have been able to address each comment as follows. With these edits, we believe the paper is now ready for publication in IJMS.
The Authors
Reviewer 1
The manuscript maintains clarity throughout; however, pivotal information concerning the lead peptide, CMP-3, seems completely absent. Additionally, a deeper investigation into the mechanisms by which CMP-3 curtails collagen fragmentation is also essential. Here are specific comments and suggestions:
- The current version of the manuscript lacks references numbered 34 through 62.
We thank the reviewer for pointing out these missing references and apologize for this oversight. We have now updated the bibliography to include a complete list of all references in the manuscript.
- A richer background on collagen mimetic peptides (CMPs) is needed. It would be beneficial to introduce their origins, offer a clear definition, and delineate their structural comparisons with collagen. Furthermore, a detailed exposition on the sequence, structure, and physicochemical attributes of CMP-3 would greatly enhance understanding.
Thank you for this brilliant suggestion that will add to the clarity and understanding of our work. We have added some background information regarding collagen and CMP structure (see lines 68-71) and have added more details about the CMP-3 compound used in our experiments to the methods section (lines 289-300). This information describes the peptide sequence and structure [single strand of (Pro-Pro-Gly)7], its synthesis and purification processes, and the final purity range (90.5-90.7%).
- The Materials section appears incomplete without a list of materials and reagents utilized. Most crucially, the source and purity level of CMP-3 should be explicitly stated.
Thank you for this suggestion. Throughout the methods section we have added additional details pertaining to each of the compounds used (including catalog numbers, see lines 245, 246, 255, 257, 279). We also elaborated on CMP-3 sequence, synthesis, source, and purity in methods section lines 289-300.
- The y-axis labeling in Figure 1B is absent.
We thank the reviewer for noticing this oversight. We have now updated the figure with the correct axis label.
- The manuscript would benefit from a clarification regarding the localization of CMP-3. Is it primarily internalized within the tissues, or does it predominantly inhabit the extracellular matrix (ECM)?
We agree with the reviewer that this would be a very useful addition to the manuscript for readers. We have made it clearer in the introduction that PPS and glial laminal tissues are tissues that contain a large amount of collagen-rich ECM (lines 74-76). This is where collagen is located within the tissues and thus where CMPs will bind due to their high affinity for damaged collagen.
- An elucidation on the mechanism of CMP-3 in attenuating collagen fragmentation is necessary. Specifically, is there any interaction between CMP-3 and MMP-1?
We thank the reviewer for this important suggestion and have added some expansion on this topic in the discussion (lines 192-197). Since CMPs are small peptide sequences directed to conserved collagen sequences, it is unlikely that CMPs would directly interact with MMPs and this has not been observed. We have therefore included some discussion on what we think the likely impact of collagen repair is by CMPs to change collagen fragmentation levels. CMP-3 has a high affinity for damaged collagen, intercalating into collagen strand breaks, repairing the native triple helical structure – i.e., reducing fragmentation directly. We also postulate that repairing collagen structure with CMPs may have important implications for downstream immune signaling which could in turn impact the expression and activation of tissue-resident MMP enzymes (to affect tissue stiffness and fragmentation of collagen).
Reviewer 2 Report
Comments and Suggestions for Authors
Interesting work, basically well written, but I have some concerns about the methodology and some other bits ( probably overlooked)
Introduction:
line 59 - all theses are mammals - so please replace "animals" with "mammals"
Materials and methods: This is a little confusing as I had to read it several time to figure out what the authors actually did.
1) How many rats did you use? (section 4.1) Start with the last sentence first. Then list the protocol. Protocol and grant numbers need to be somewhere in the manuscript ( materials or in the acknowledgements)
2) Exactly how many rats per treatment? Figure 1 and 2 cite n=1, which is not a acceptable level of replication (minimum is 3per treatment).
3) I understand AFM first (is this the baseline referred to in t figures 1 and 2?), MMP-1 (degradation - any IHC then?) and the either CMP-3 (restore + IHC) or PBS (control + IHC). This is the crux of experimental design and needs to be explained more prominently - in the introduction (without the foreshadowing of the results). Did you do IHC on undegraded fixed specimens? What was your control?
Results
line 82: now you mention distinct PPS locations? Not in methods (or I missed that). Perhaps a little diagram showing PPS vs. GL locations?
Discussion
At some stage there does need to be an explanation on how CMP-3 is thought to work!
No concluding remarks with questions that arose from this work
Author Response
We thank the reviewers of our manuscript for their thoughtful and comprehensive comments. We are delighted to submit our revision, in which we have been able to address each comment as follows. With these edits, we believe the paper is now ready for publication in IJMS.
The Authors
Interesting work, basically well written, but I have some concerns about the methodology …
Introduction:
line 59 - all theses are mammals - so please replace "animals" with "mammals"
Materials and methods: This is a little confusing as I had to read it several time to figure out what the authors actually did.
Thank you for this suggestion. We have replaced the word “species” with “mammals” in line 59. To avoid confusion regarding experimental design, we have also added a flow chart (Figure 5, line 300) clearly demonstrating the sequence of measurements and treatments.
- How many rats did you use? (section 4.1) Start with the last sentence first. Then list the protocol. Protocol and grant numbers need to be somewhere in the manuscript ( materials or in the acknowledgements)
Thank you for this suggestion which will improve the clarity of our manuscript. We edited section 4.1. so that it starts with the last sentence. Although we mentioned the number of rats in our figure legends, we agree that mentioning it again in the methods section will improve the clarity of our experimental design. Therefore, we clarified the animal number “(n=6; 3 for PPS measurements and 3 for glial lamina measurements)” in lines 230 and 231.
- Exactly how many rats per treatment? Figure 1 and 2 cite n=1, which is not a acceptable level of replication (minimum is 3 per treatment).
Thank you for this comment. Although the legends of figures 1C and 2C and tables 1 and 2 also cite n=3 for the pooled data analysis, we also edited the methods section by clearly stating the animal number “(n=6; 3 for PPS measurements and 3 for glial lamina measurements)” in lines 230 and 231. The data shown in figures 1 and 2 are representative measurements from one location in one rat sample and we have made this clear stating “representative data n=1 of 3 animals).
- I understand AFM first (is this the baseline referred to in t figures 1 and 2?), MMP-1 (degradation - any IHC then?) and the either CMP-3 (restore + IHC) or PBS (control + IHC). This is the crux of experimental design and needs to be explained more prominently - in the introduction (without the foreshadowing of the results).
Thank you for this excellent suggestion to improve clarity. We have added a sentence outlining the experimental design in lines 80-82 of the introduction. We also added a new figure (Figure 5) to clearly demonstrate the experimental flow.
- Did you do IHC on undegraded fixed specimens? What was your control?
We performed IHC on naïve sections, i.e., tissue not degraded by MMP-1 (naïve images in figure 3B; 1a and 2a) to show levels of fragmented collagen occurring in the native tissue state. We then performed IHC on AFM experimental tissue that had been digested with MMP-1 then treated with PBS (vehicle control) or digested with MMP-1 and treated with CMP (experimental). This is now clarified in the manuscript.
Results
line 82: now you mention distinct PPS locations? Not in methods (or I missed that). Perhaps a little diagram showing PPS vs. GL locations?
Thank you for this excellent suggestion. We have edited the methods section by adding “distinct” in lines 264 and 266 to clarify that we have acquired data from multiple distinct peripapillary scleral and glial laminal locations in each sample. These separate or “distinct” locations of measurements in 10um x 10um scan sized zones fall within the regions of interests (ROI) depicted on Figure 4B.
Discussion
At some stage there does need to be an explanation on how CMP-3 is thought to work!
No concluding remarks with questions that arose from this work
We thank the reviewer for pointing out this important point. We have edited the discussion to describe the possible effect of CMP on the level of collagen fragmentation in tissue and its downstream implications (lines 192-197) and have added to the last paragraph of the discussion to clearly state our conclusion and suggest future investigations that would build on our current work (lines 219-221). Please see the response to reviewer 1’s similar suggestion.
Reviewer 3 Report
Comments and Suggestions for Authors
This paper examines the ability of collagen mimetic peptides to repair the types of collagen present in ocular structures after exposure to MMPs. Eyes were surgically removed from rats and exposed to MMP-1 and then treated with CMP-3 or vehicle. The integrity and quality of the collagen in ocular structures was then examined using Atomic Force Microscopy and immunohistochemistry. The CMP-3 treatment showed significantly reduced collagen fragmentation and overall increased integrity and structure.
The paper is overall straightforward with a sound study design and clear presentation of results. There are many questions remaining such as immune response or the quality of vision post-treatment, which would be interesting for further study in the future. I believe this paper is a good contribution to our understanding of injury to and possible repair of ocular structures and is suitable for publication in its current state.
Author Response
We thank the reviewers of our manuscript for their thoughtful and comprehensive comments. We are delighted to submit our revision, in which we have been able to address each comment as follows. With these edits, we believe the paper is now ready for publication in IJMS.
The Authors
… The paper is overall straightforward with a sound study design and clear presentation of results. There are many questions remaining such as immune response or the quality of vision post-treatment, which would be interesting for further study in the future. I believe this paper is a good contribution to our understanding of injury to and possible repair of ocular structures and is suitable for publication in its current state.
We thank the reviewer for their kind and supportive review of our work. We agree that investigating the effect of CMP repair on tissue immune responses and visual acuity in animal models of ocular diseases is a fundamental next step and now mention this in the discussion (lines 223-227).
Round 2
Reviewer 1 Report
Comments and Suggestions for Authors
The authors have carefully addressed my comments.